# Selective killing of homologous recombination-deficient cancer cell lines by inhibitors of the RPA:RAD52 protein-protein interaction

**Mona Al-Mugotir, Jeffrey J. Lovelace, Joseph George, Mika Bessho, Dhananjaya Pal, Lucas Struble, Carol Kolar, Sandeep Rana, Amarnath Natarajan, Tadayoshi Bessho\*, Gloria E. O. Borgstahl\***

The Eppley Institute for Research in Cancer and Allied Diseases, Fred & Pamela Buffett Cancer Center, University of Nebraska Medical Center, Omaha, Nebraska, United States of America

\* gborgstahl@unmc.edu (GEOB); tbessho@unmc.edu (TB)

**Data Availability Statement:** All relevant data are within the manuscript and its Supporting information files.

## Abstract

Synthetic lethality is a successful strategy employed to develop selective chemotherapeutics against cancer cells. Inactivation of RAD52 is synthetically lethal to homologous recombination (HR) deficient cancer cell lines. Replication protein A (RPA) recruits RAD52 to repair sites, and the formation of this protein-protein complex is critical for RAD52 activity. To discover small molecules that inhibit the RPA:RAD52 protein-protein interaction (PPI), we screened chemical libraries with our newly developed Fluorescence-based protein-protein Interaction Assay (FluorIA). Eleven compounds were identified, including FDA-approved drugs (quinacrine, mitoxantrone, and doxorubicin). The FluorIA was used to rank the compounds by their ability to inhibit the RPA:RAD52 PPI and showed mitoxantrone and doxorubicin to be the most effective. Initial studies using the three FDA-approved drugs showed selective killing of BRCA1-mutated breast cancer cells (HCC1937), BRCA2-mutated ovarian cancer cells (PE01), and BRCA1-mutated ovarian cancer cells (UWB1.289). It was noteworthy that selective killing was seen in cells known to be resistant to PARP inhibitors (HCC1937 and UWB1 SYr13). A cell-based double-strand break (DSB) repair assay indicated that mitoxantrone significantly suppressed RAD52-dependent single-strand annealing (SSA) and mitoxantrone treatment disrupted the RPA:RAD52 PPI in cells. Furthermore, mitoxantrone reduced radiation-induced foci-formation of RAD52 with no significant activity against RAD51 foci formation. The results indicate that the RPA:RAD52 PPI could be a therapeutic target for HR-deficient cancers. These data also suggest that RAD52 is one of the targets of mitoxantrone and related compounds.

## Introduction

Cancer cells often show dependence on unique DNA repair pathways due to frequent mutations and higher demand for DNA damage repair to sustain their active division. In cancer

 

**Funding:** The research was supported by the Fred & Pamela Buffett Cancer Center Support Grant (P30CA036727) pilot project funding (GB & TB), the Nebraska Research Initiative (GB), the Nebraska Department of Health and Human Services (2015, 2017-08, 2018-11; GB); and the U. S. Department of Education GAANN (P200A120231) (GB) and Nebraska NASA EPSCoR Space Grant (MAM) for student fellowships.

**Competing interests:** The authors have declared that no competing interests exist.

therapy design, it is desirable to target factors that are dispensable for normal cells, thus sparing them from collateral damage [1]. Coupling maladaptive mutations in cancer with pharmacological inactivation of another target to induce selective cell death is called synthetic lethality [2]. This approach promises minimization of side effects that exhaust patients and has the potential to lift the roadblocks for the treatment of aggressive subtypes of cancers for which there are currently no effective therapies. Ovarian high-grade serous carcinoma (HGSC) and triple-negative breast cancer (TNBC), for example, are among the most lethal subtypes characterized by a poor prognosis, high rate of reoccurrence and metastasis [3, 4]. Many patients with these specific cancer subtypes harbor somatic mutations in *BRCA1* or *BRCA2*, or other homologous recombination (HR) genes, which inspired the development of targeted therapeutics [3–7]. Poly(ADP-ribose) polymerases (PARP) catalyze the polymerization of ADP-ribose moieties and facilitate base excision repair (BER) and single-strand break (SSB) repair. PARP inhibitors (PARPi) impair SSB repair and cause the formation of DSBs [8]. PARPi also inhibits the dissociation of PARP from DNA damage sites, and the trapped PARP induces replication-dependent DSBs. Cancer cells with defects in HR repair show synthetic lethality with PARP and are selectively vulnerable to DSBs generated by PARPi [9]. Despite the promising successes of PARPi in pre-clinical settings, in clinical trials, patients commonly acquire resistance to PARPi [10].

Since the discovery of synthetic lethality of PARPi with BRCA1/2-deficient cancer cell lines, there have been many studies to discover other synthetic lethality interactions related to DNA repair [11, 12]. The deletion of RAD52 was discovered to be synthetically lethal with deficiencies in HR repair [13, 14]. RAD52 participates in HR repair and is conserved throughout eukaryotes. In mammalian cells, RAD52 deletion does not compromise HR repair; however, the synthetic lethality of RAD52 inactivation with a BRCA-mediated HR deficiency indicates a critical role for RAD52-mediated DNA repair pathway(s) in the absence of BRCA-mediated HR repair [15]. Recent studies showed that RAD52 is involved in multiple DNA repair pathways, including a BRCA-independent HR repair involving RAD51 [16], single-strand annealing (SSA) [17], break-induced replication (BIR) repair [18, 19], RNA-templated double-strand break (DSB) repair [20, 21], and transcription-associated HR involving XPG [22]. However, how RAD52 activity helps HR-deficient cancers to survive is not clear.

Several studies have explored targeting RAD52 to attack cancer cells with some success. MicroRNA-302 was found to reduce the expression of AKT1 and RAD52, making breast cancer cells more sensitive to radiation [23]. A peptide F79 aptamer targeting the RAD52 DNA binding domain induced synthetic lethality in HR-deficient leukemia without affecting normal cells and enhanced the effect of already-approved drugs [24]. The small molecule 6-hydroxy-DL-dopa (6-OH-dopa) was identified as an allosteric inhibitor of the RAD52 ssDNA binding domain with an $IC_{50}$ of 1.1 μM, but unfortunately, it is also neurotoxic and induces Parkinson's disease [25]. Interestingly, both F79 peptide and 6-OH-dopa break up RAD52 rings in agreement with our earlier results [26]. Other RAD52 small molecule inhibitors (SMIs) were identified from screens based on annealing activity and by docking to the ssDNA binding site. Of these, the compounds D-G23, D-I03, AICAR, A5'MP, epigallocatechin, and epigallocatechin-3-monogallate appear to be the most effective [27–29].

In this study, we took a different approach and targeted the RPA:RAD52 protein-protein interaction (PPI). RAD52 contains several interaction domains [30, 31]. The self-association domain in the N-terminal half of RAD52 is responsible for the formation of a thermally-stable ring of monomers with residues 160 to 209 reaching over and hugging the adjacent monomer [31–33]. Each RAD52 monomer binds four nucleotides, and two ssDNA binding sites were found–one in a positively-charged groove and the other on the outside of the ring [34, 35]. Biophysical data showed that RAD52 is a monomer when bound to RPA [26]. There are two

domains on RAD52 involved in RPA binding that correspond to two domains on RPA that bind RAD52 [36]. These PPIs and ssDNA binding sites could be essential for RAD52 DNA repair activities. In the process of repairing a DSB, ssDNA tails are enzymatically created and bound by RPA. Then RAD52 comes along and binds RPA before binding ssDNA. There is a hand-off of ssDNA from RPA to RAD52. In fact, the RPA:RAD52 complex is required for RAD52 foci formation [37] at DSBs and is required for SSA of longer plasmid DNA [38].

In this report, we conducted a high throuput screen (HTS) for SMIs targeting RPA: RAD52 PPI using our newly developed Fluorescence-based protein-protein Interaction Assay (FluorIA) [39]. Over 101,500 compounds were screened from three different chemical libraries. Eleven candidate hits were obtained, five of which showed strong inhibition of RPA: RAD52 PPI as determined by their low half-maximal, effective concentration ($EC_{50}$) values *in vitro*. Three hits were the following FDA-approved drugs: quinacrine, mitoxantrone, and doxorubicin. Cancer cell lines lacking BRCA1 or BRCA2 functions demonstrated selective sensitivity to quinacrine, mitoxantrone, and doxorubicin. Furthermore, mitoxantrone inhibited SSA, radiation-induced RAD52 foci formation and the RPA:RAD52 PPI in cells. These early results indicate that inhibition of RPA:RAD52 PPI appears to inhibit RAD52-mediated DNA repair and that mitoxantrone may be a potent RAD52 inhibitor.

## Results

### High throughput screening to identify SMIs of RPA:RAD52 PPI with FluorIA

To identify SMIs of the RPA:RAD52 PPI, three chemical libraries were screened using FluorIA (Table 1). No hits were obtained from SelleckChem Kinase Inhibitor Library, but three hits came from the Prestwick library, and eight hits were obtained from the Chembridge library. In total, eleven hits were found (Fig 1A), and their ability to inhibit the RPA:RAD52 PPI was ranked using FluorIA (Fig 1B). Interestingly, FDA-approved drugs, mitoxantrone, and doxorubicin were the most potent inhibitors in FlourIA ($EC_{50}$ values of 29.7 and 10.1 μM, respectively; Table 2). The Chembridge hit 5316833 came in third place with an $EC_{50}$ of 66 μM.

### Selective cytotoxicity of quinacrine, mitoxantrone, and doxorubicin in HR-deficient cancer cell lines

We were curious to see if RPA:RAD52 PPI inhibitors would show synthetic lethality in HR-deficient cancer cell lines, so we examined the cytotoxicity of quinacrine, mitoxantrone, and doxorubicin in cells with impaired *BRCA1* or *BRCA2* function. Following the 72-hour treatment with indicated concentrations of each compound, quinacrine mitoxantrone, or doxorubicin, the viability of each cell line was analyzed. The $EC_{50}$ values of each compound in each cell line tested are indicated in the graphs. Mitoxantrone and doxorubicin preferentially killed the HR-deficient cancer cell lines, HCC1937, UWB1.289, and PE01 compared to their HR-proficient counterparts, HCC1937+BRCA1, UWB1.289+BRCA1, and PE01C4-2, respectively (Figs 2–4). Quinacrine was similar to mitoxantrone in selectively killing HCC1937 cells (Fig 2)

**Table 1. Chemical libraries screened by FluorIA and hits obtained.**

| Chemical Library | Hits |
|---|---|
| **SellekChem** (355) | none |
| **Prestwick** (1200) | Three |
| **Chembridge** (100,000) | Eight |

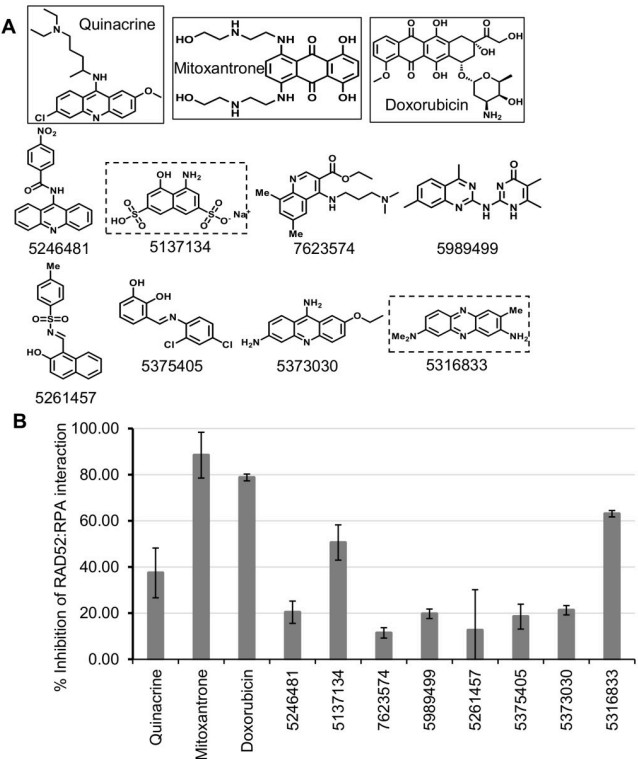

**Fig 1. Identification and characterization of RPA:RAD52 inhibitors. (A)** Chemical structures of the top eleven FluorIA hits. The strongest inhibitors are boxed (FDA-approved in solid-line, Chembridge dashed-line). **(B)** Comparison of hits for the inhibition of the RPA:RAD52 interaction at a single concentration (62.5 μM). The FluorIA inhibition data for the full concentration range for each hit was included in S1 Fig.

but killed the HR-deficient ovarian cell lines with less selectivity (Figs 3 and 4). The detection of cleaved PARP by western blotting confirmed that mitoxantrone preferentially induced apoptosis in the BRCA1-mutated UWB1.289 compared to the BRCA1-restored UWB1.289 (Fig 5).

The inhibition of PARP is also synthetically lethal to HR-deficient cancer cell lines. PARPi are effective against various BRCA-mutated cancer cell lines. However, limited success has been reported in clinical trials of PARPi with BRCA-deficient tumors. HCC1937 is reported to be insensitive to PARPi despite its BRCA1-mutated status [40–43]. Interestingly, all three compounds showed selective cytotoxicity to HCC1937 (Fig 2), indicating the RPA:RAD52 PPI is an effective target for selective killing of BRCA-mutated cancer cell line regardless of the status of synthetic lethality with PARPi. To generalize this concept, we examined the cytotoxicity of these compounds to two PARPi resistant cell lines, UWB1-SYr12 and SYr13 (Fig 3). UWB1-

**Table 2. EC$_{50}$ values for the strongest five inhibitors in FluorIA.**

| Compound | Quinacrine | Mitoxantrone | Doxorubicin | 5137134 | 5316833 |
|---|---|---|---|---|---|
| EC$_{50}$ (μM) | 97.7 | 29.7 | 10.1 | 104.7 | 66.2 |

EC$_{50}$ values obtained from a polynomial curve fitting for each compound where the response (inhibition) is tested at 15 different dose concentrations (0.03–500 μM) in triplicate. RAD52(1–303) was used as positive control. The FluorIA inhibition data for the full concentration range for each hit was included in S1 Fig.

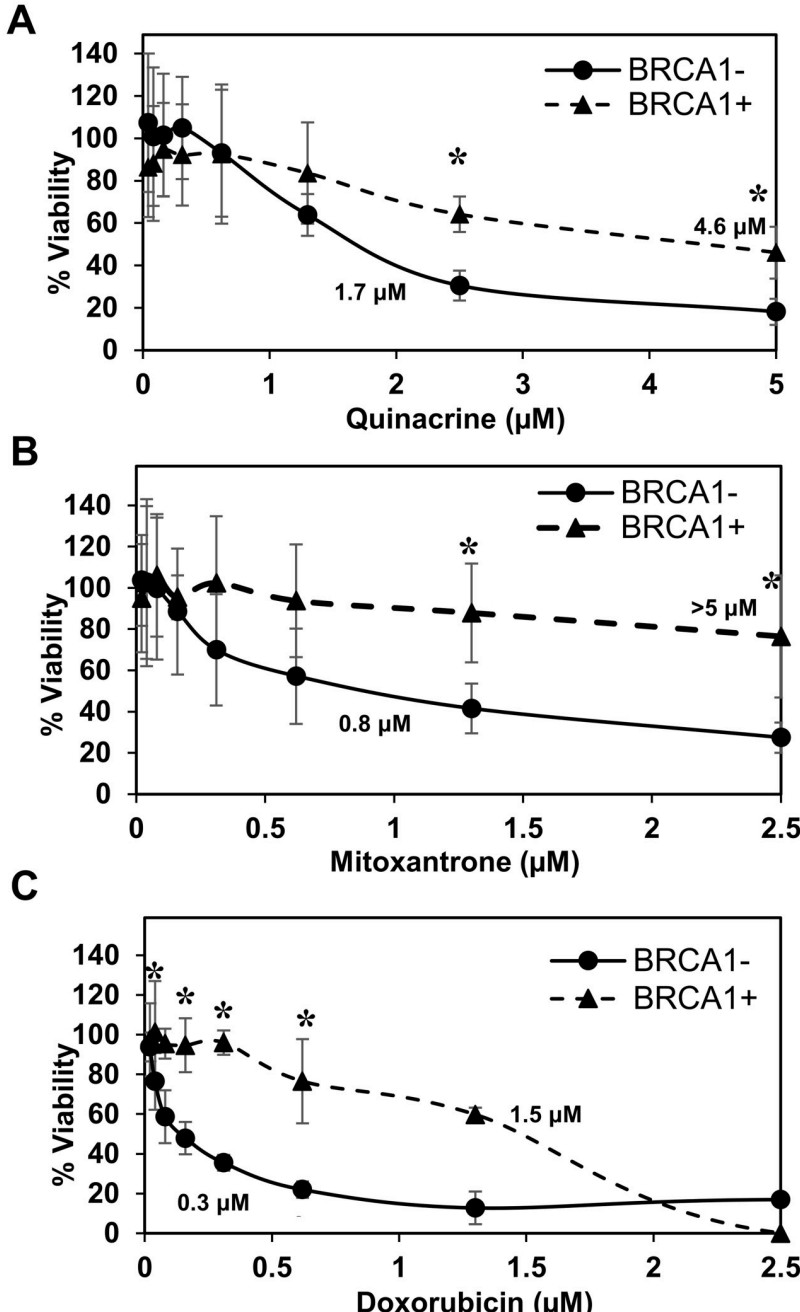

**Fig 2. BRCA1-deficient HCC1937 breast cancer cell line survival assay.** BRCA1-deficient HCC1937 cells corrected with wild-type BRCA1 gene (BRCA1+; closed triangle), and with an empty vector (BRCA1-; closed circle) were examined. For each experiment (Figs 2–4), cell lines were treated with nine concentrations of (**A**) quinacrine, (**B**) mitoxantrone, and (**C**) doxorubicin in 96-well culture plates at $5x10^3$ cell/well density. Each treatment point was made in sextuplicate. Treated cells were incubated for 72 hours at 37°C before assessing cell viability using PrestoBlue. Data were normalized to vehicle control. The experiment was repeated three times. Error bars indicate the standard deviation. Significantly different data are indicated with an asterisk. $EC_{50}$ values, estimated from the graphs, are given.

SYr12 and SYr13 are laboratory-generated PARPi-resistant cell lines derived from BRCA1-mutated UWB1.289. These two cell lines gained the BRCA1-independent, ATR-dependent loading of RAD51 in HR and fork protection and restored PARPi resistance [43]. Mitoxantrone and doxorubicin showed preferential cytotoxicity to the PARPi-resistant UWB1-SYr12

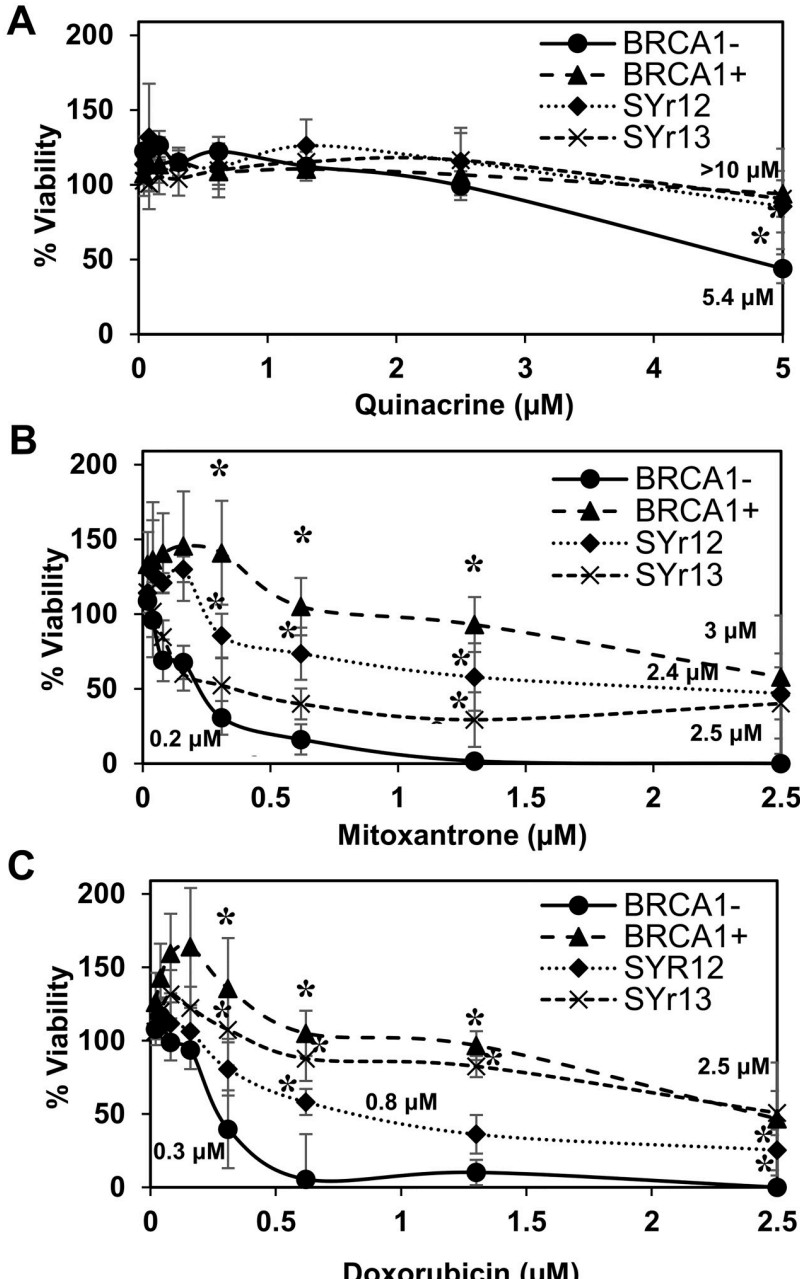

**Fig 3. BRCA1-deficient UWB1.289 ovarian cancer cell line survival assay.** BRCA1-deficient UWB1.289 cells corrected with wild-type BRCA1 gene (BRCA1+; closed triangle) and parental BRCA1-deficient UWB1.289 (BRCA1-; closed circle). Two PARPi-resistant cell lines derived from UWB1.289, UWB1.289.SYr12 (closed diamond) and UWB1.SYr13 (X), were also examined. Cell lines were treated with (**A**) quinacrine, (**B**) mitoxantrone, and (**C**) doxorubicin as described in Fig 2.

and SYr13, although their selectivity was moderate compared to the one with parental UWB1.289 (Fig 3).

The level of RAD52 protein and DNA damage before and after mitoxantrone treatment was characterized and was found to vary between cell lines (Fig 6). For HCC1937 and UWB1 cell lines, low to negligible levels of DNA damage occurred from mitoxantrone treatment and

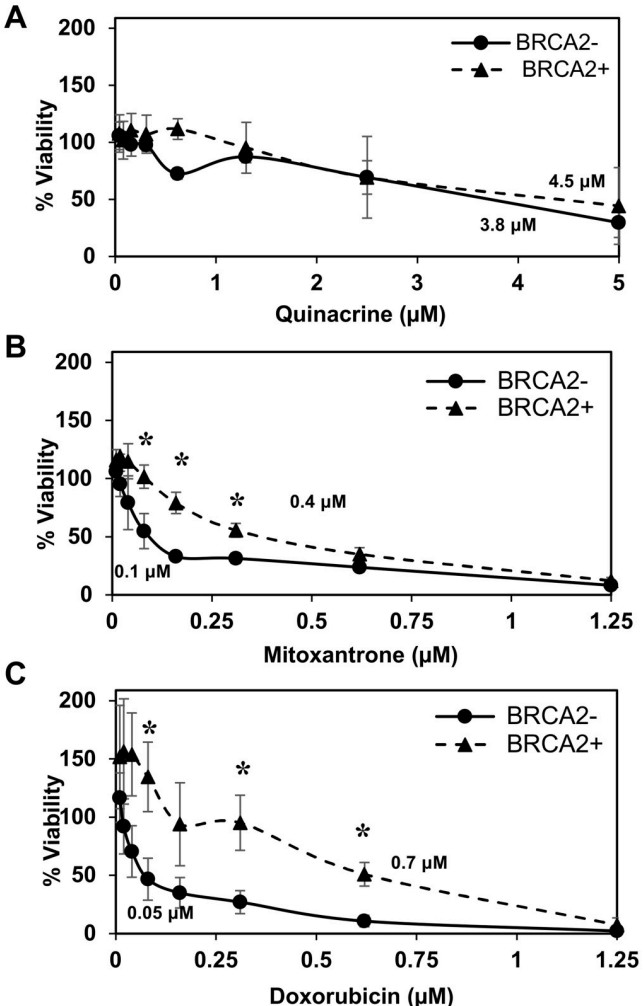

**Fig 4. BRCA2-deficient PE01 ovarian cancer cell line survival assay.** BRCA2-deficient PE01(BRCA2-; closed circle) and BRCA2-revertant PE01 C4-2 (BRCA2+; closed triangle) were examined. Cell lines were treated with (**A**) quinacrine, (**B**) mitoxantrone, and (**C**) doxorubicin as described in Fig 2.

RAD52 levels remained relatively high. In HCC1937 cells the level of RAD52 increased with mitoxantrone treatment and in UWB1 cells the level of RAD52 decreased. Interestingly, the PE01 ovarian cancer cell line was more sensitive to mitoxantrone treatment showing a dramatic reduction in RAD52 protein and a substantial increase in DNA damage, relative to the other cell lines.

## Inhibition of DNA repair activities of RAD52 by mitoxantrone

In mammalian cells, RAD52 promotes SSA but has little impact on HR. Because SSA is RAD52-mediated, but independent of BRCA1, BRCA2, and RAD51, SSA has been implicated as one of the backup DSB repair pathways in the absence of the BRCA-mediated HR repair. To examine the impact of the inhibitors of the RPA:RAD52 PPI on SSA, we employed a cell-based SSA assay. Pre-treatment of U2OS-SA and U2OS-DR cells with mitoxantrone revealed that mitoxantrone suppressed both SSA and HR (Fig 7A). Mitoxantrone preferentially suppressed the SSA activity over the HR activity at 4 nM. SSA was inhibited by 72% ± 7%, and HR

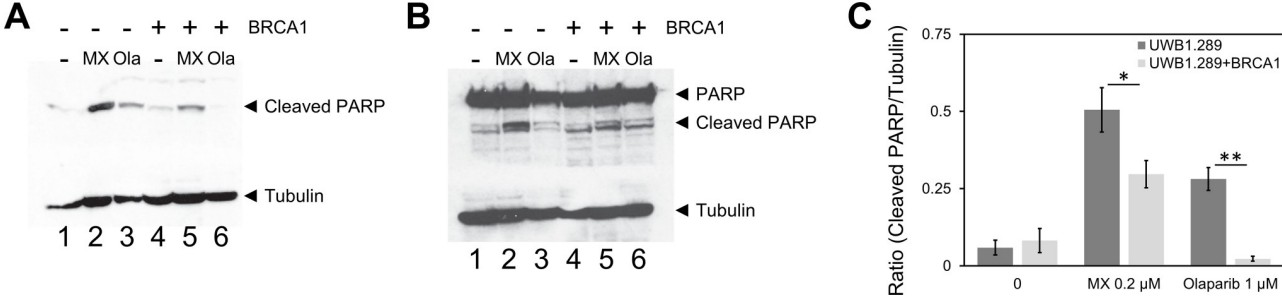

**Fig 5. Mitoxantrone induces apoptosis preferentially in the BRCA1-deficient UWB1.289.** UWB1.289 cells with or without restoration of the BRCA1 expression were treated with 0.2 μM of mitoxantrone (MX) for three days. Olaparib (Ola) at 1 μM was used as control for the induction of apoptosis. Near $EC_{50}$ concentrations were selected for each compound. After the treatment with each compound, cell lysates were prepared and cleaved PARP was detected with **(A)** anti-cleaved PARP antibody (Cell Signaling Technology #9541) and **(B)** anti-PARP antibody (Cell Signaling Technology #9542). Tubulin detected with anti-beta-tubulin antibody (Cell Signaling Technology #15115) was used as an internal control. **(C)** Ratios of cleaved PARP to tubulin were determined with ImageJ and graphed. Three independent experiments were performed and the mean values are shown. The bars indicate standard deviations. Statistical significance of differences was determined with Student t-test (* $p < 0.05$ and ** $p < 0.01$).

was suppressed by 40% ± 7% at 4 nM (Fig 7B). With higher concentrations of mitoxantrone (> 6 nM), both SSA and HR were inhibited at similar levels, partly due to the suppression of the exogenous expression of I-SceI (Fig 7C). The cell-based DSB repair assays used in this study induce DSBs by the exogenous expression of the I-SceI endonuclease by transfection. Because mitoxantrone can inhibit transcription, we measured the expression of I-SceI and GFP by western blot analysis. DSB repair activity measured by Western blot analysis also confirmed the preferential inhibition of SSA over HR by mitoxantrone (Fig 7D and 7E).

RAD51 forms foci after X-ray radiation. The radiation-induced RAD51 foci are considered to be a surrogate marker for HR activity. RAD52 also forms foci after irradiation and the foci formation is thought to represent RAD52-mediated DNA repair processes. GFP-RAD52 and GFP-RAD51 were expressed individually in HR-restored PE01 C4-2, and the impact of mitoxantrone on radiation-induced foci formation of GFP-RAD52 and GFP-RAD51 were examined. As expected, GFP-RAD52 and GFP-RAD51 formed foci after radiation in the HR-proficient PE01 C4-2 (Fig 8, left panel). Pre-treatment of PE01 C4-2 with mitoxantrone (3 nM) inhibited the radiation-induced RAD52 foci formation (Fig 8, right top) while it did not suppress the radiation-induced RAD51 foci formation (Fig 8, right bottom). Thus, mitoxantrone appears to inhibit RAD52-mediated DNA repair pathways, including SSA but has little impact on the RAD51-mediated HR. These data suggest that the inhibitors of the RPA:RAD52 PPI can selectively suppress RAD52-based DNA repair without altering the RAD51-based DNA repair significantly.

## Discussion

In this report, we aimed to compromise RAD52 activity by inhibiting its PPI with RPA. Our newly developed FluorIA is a PPI assay that was optimized to use in high throughput settings to screen SMIs. We successfully identified three FDA-approved drugs as the inhibitor of the RPA:RAD52 PPI. Mitoxantrone and doxorubicin, are anti-cancer drugs in clinical use and displayed RPA:RAD52 PPI inhibition in FluorIA and had the highest cytotoxicity in BRCA-deficient cancer cell lines. BRCA1 and BRCA2 cancer cell lines also demonstrated more limited sensitivity to quinacrine. Furthermore, our results showed that mitoxantrone suppressed SSA preferentially over HR repair. Mitoxantrone also inhibited X-ray induced RAD52 foci formation without affecting RAD51 foci formation. It is worth noting that experiments employing size exclusion chromatography followed by the co-immunoprecipitation of RAD52 with RPA

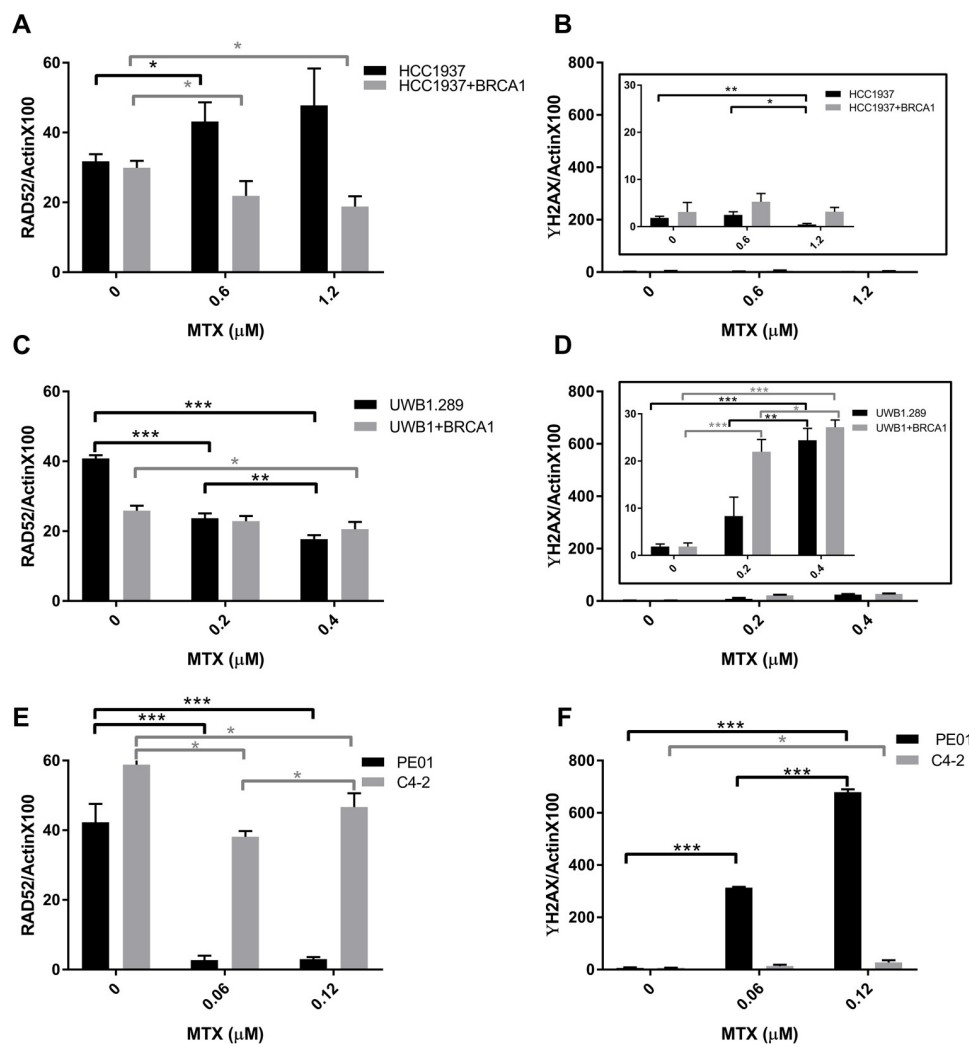

**Fig 6. The effect of mitoxantrone treatment on RAD52 levels and DNA damage. (A, C, and E)** RAD52 and γH2AX **(B, D, and F)** levels were probed in the three parental (black) and (BRCA1 or BRCA2 proficient, grey) cell lines indicated was assessed with Simple Western using lysate post 72 h treatment with mitoxantrone at doses near their EC$_{50}$ values (see Figs 2–4). Data and ANOVA statistical analyses are described in the material and methods section. Significance statistics are shown with single, double, or triple asterisks representing P<0.05, P<0.005, or P<0.0005, respectively.

indicate that mitoxantrone disrupts the RPA:RAD52 PPI in cells (Fig 9). Interestingly, preliminary experiments show that incubation with 3 nM mitoxantrone disrupted the interaction of RPA with high molecular weight rings of RAD52, and only the interaction with a form of RAD52 near 66 kDa was retained. Overall these results confirm that the RPA:RAD52 PPI is crucial for RAD52-mediated DNA repair pathways, as was previously reported [37]. In the absence of the BRCA-mediated HR repair, cells can become addicted to RAD52-based DNA repair pathways to survive. The inhibition of the interaction abrogates RAD52-mediated DNA repair pathways selectively and leads to cell death. Since RAD52 is involved in various DNA repair pathways, it is unclear which RAD52-mediated DNA repair pathway keeps the cells alive in the absence of BRCA-mediated HR. Although identification of DNA repair factors involved in each RAD52-mediated DNA repair pathway is required to reveal the mechanistic basis of the synthetic lethality of HR-deficient cancers with RAD52 inhibition, our results

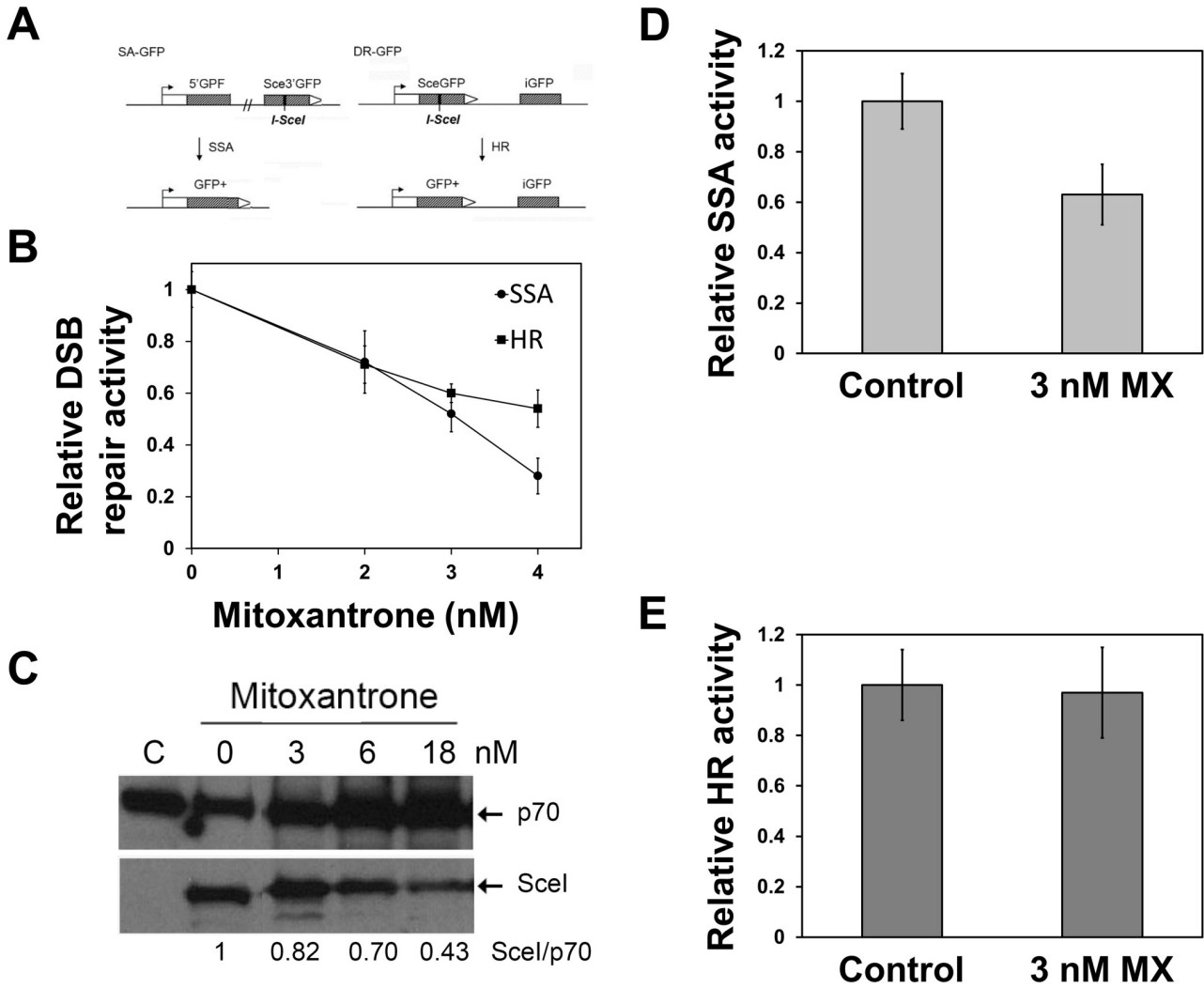

**Fig 7. RAD52 inhibitor mitoxantrone selectively suppresses SSA. (A)** The effect of mitoxantrone (MX) on SSA and HR were studied with cell-based DSB repair assays. U2OS-SA and U2OS-DR cells were pre-treated with mitoxantrone for two hours before transfection of the SceI-expression plasmid. Then cells were grown in the presence of mitoxantrone for three days. DSB repair activities were determined by **(B)** FACS analysis. FACS data are provided in S3 Fig. **(C)** Inhibition of the expression of I-SceI (SceI) by mitoxantrone. U2OS-SA cells were pre-treated with indicated concentrations of mitoxantrone for two hours, before I-SceI-expression vector transfection. After 72 hours incubation, the cells were harvested, and lysates were prepared. Lysate from each treatment was analyzed by 12% SDS-PAGE and the expression levels of I-SceI was determined by the western blot using anti-HA antibody. The p70 of RPA was used as a loading control. Ratios of I-SceI to p70 was obtained with ImageJ and are shown as the bottom of the image. C: control experiment without the addition of mitoxantrone. **(D, E)** Analysis of DSB repair activiies by western blots. Graphs showed relative DSB repair activities with mitoxantrone to control experiments without mitoxantrone. At least three independent experiments were performed. Error bars represent standard deviations. The differences between the SSA inhibition and the HR suppression by mitoxantrone were statistically significant ($p<0.05$ at 4 nM, and $p<0.05$ at 3 nM). Western blot data are provided in S4 Fig.

demonstrated that the RPA:RAD52 PPI is highly likely to be essential for RAD52 addiction in BRCA-deficient cancers.

So far, mitoxantrone and doxorubicin are the best at inhibiting the RPA:RAD52 PPI, inhibiting RPA:RAD52 activity, and at selective killing. Mitoxantrone is among a class of chemotherapeutic drugs with flat polycyclic aromatic systems that can intercalate DNA [44, 45], and mitoxantrone is known to inhibit topoisomerase IIα and to create DSBs [46]. Mitoxantrone specifically targets topoisomerase IIα isoform, and this gives a safety advantage in patients because topoisomerase IIα is preferentially expressed in proliferating cells while topoisomerase

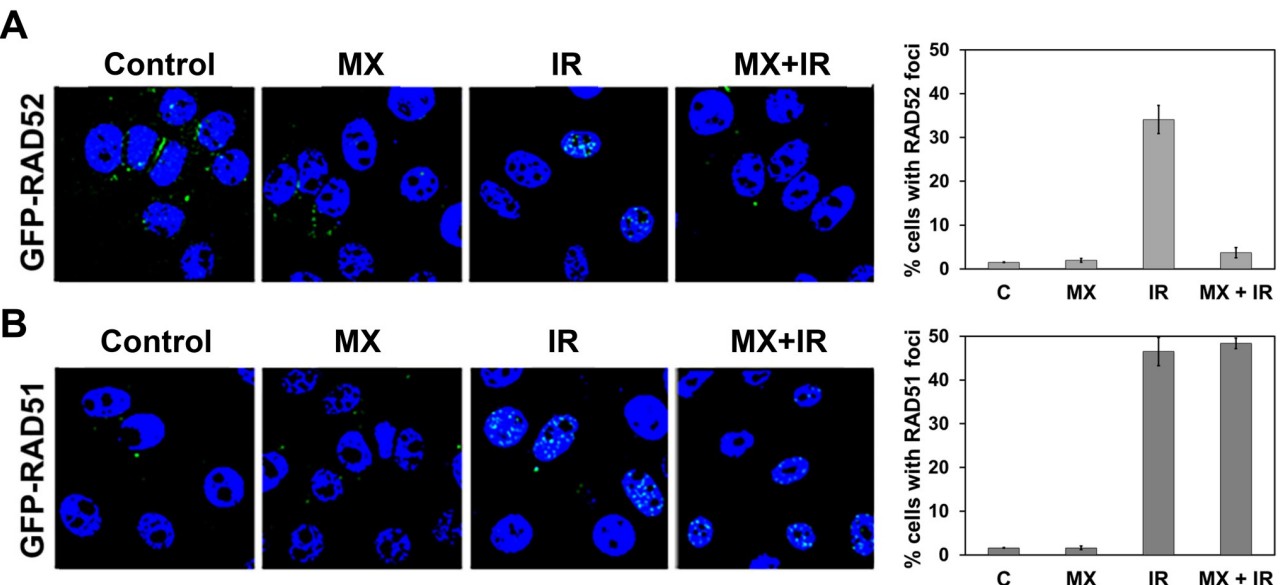

**Fig 8. Effect of RAD52 inhibitor mitoxantrone on radiation-induced GFP-RAD52 and GFP-RAD51 foci formation in the HR-proficient ovarian cancer cells.** PE01 C4-2 cells constitutively expressing **(A)** GFP-RAD52 and **(B)** GFP-RAD51 were treated with mitoxantrone (MX, 3 nM) alone, radiation alone (IR, 15 Gy, RAD Source RS2000) or both. Cells without mitoxantrone and radiation (labeled C) were controls. The average percent of cells with foci (more than 5 foci per cell) are graphed from three independent experiments. More than 30 cells per experiment were analyzed. Error bars represent standard deviation. GFP-RAD52 and GFP-RAD51 were stably expressed individually in PE01 C4-2 (S5 Fig).

IIβ is expressed by quiescent cardiomyocytes [47]. Doxorubicin, on the other hand, targets both isoforms contributing to greater cardiotoxic risk and limited dose tolerance [48, 49]. Tolerance for high doses of mitoxantrone as part of multi-agent chemotherapy and better efficacy was reported in acute myeloid leukemia (AML) [50]. These properties of mitoxantrone are beneficial to patients, although still not without serious side effects. There is room for improvement.

A few newer studies support the notion that the action of mitoxantrone involves more than just the creation of DSBs. MTX appears to induce apoptosis in osteosarcoma cells through the regulation of the Akt/FOXO3 pathway [51]. Mitoxantrone inhibits PknB, PIM1, and protein kinase C as a nanomolar ATP competitive inhibitor [52–54]. Mitoxantrone also activates T-cell protein tyrosine phosphatase [55] and can inhibit VIM2 [56]. So mitoxantrone's effects are likely attributed to multiple targets that are vital to cell survival [51, 53]. Mitoxantrone treated cells have also shown some changes to gene expression profiles [57, 58]. Our western analysis before and after mitoxantrone treatment show that in PEO1 cells mitoxantrone reduces RAD52 protein levels significantly, there is a significant increase in DNA damage and this synthetic lethality is part of the mechanism for the low EC$_{50}$ of 0.1 μM in PEO1 cells (Fig 6E and 6F). This reduction in RAD52 levels is less significant in the C4-2 cell line that is a clone derived from PE01 that has a secondary mutation that restored BRCA2 function and there are low levels of DNA damage. These effects were not as significant in the other two HR-deficient cell lines (Fig 6A–6D). The mechanism of death in HCC1937 and UWB1 does not appear to rely as heavily on DNA damage from mitoxantrone and RAD52 levels increase or decrease but remain high. The exact mechanism for how mitoxantrone alters RAD52 protein levels, specifically or not, is not known.

While there have been a few reports about quinacrine as an anti-cancer agent, there is limited knowledge regarding the mechanism of its action against cancer. Quinacrine has been

**A** Control

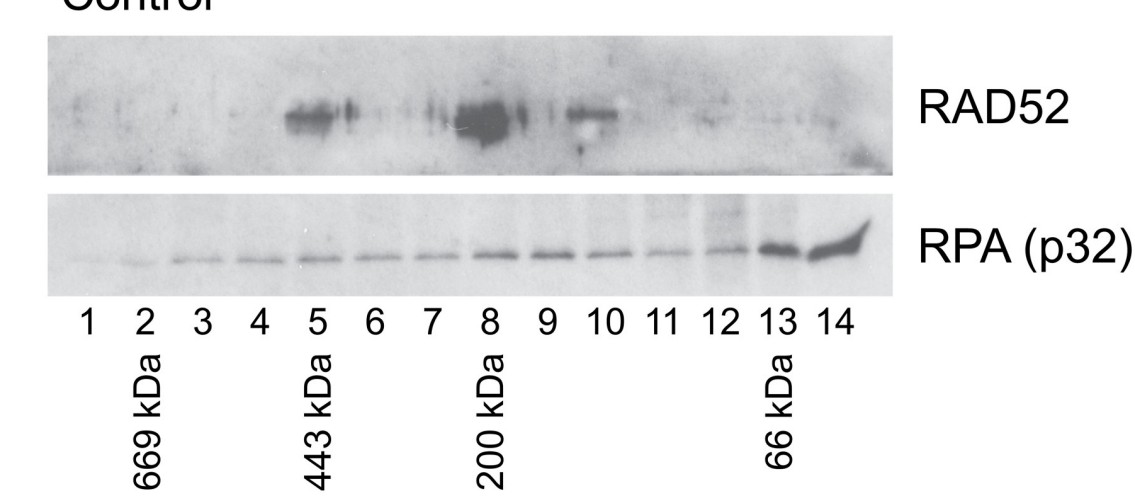

**B** MTX at 3 nM

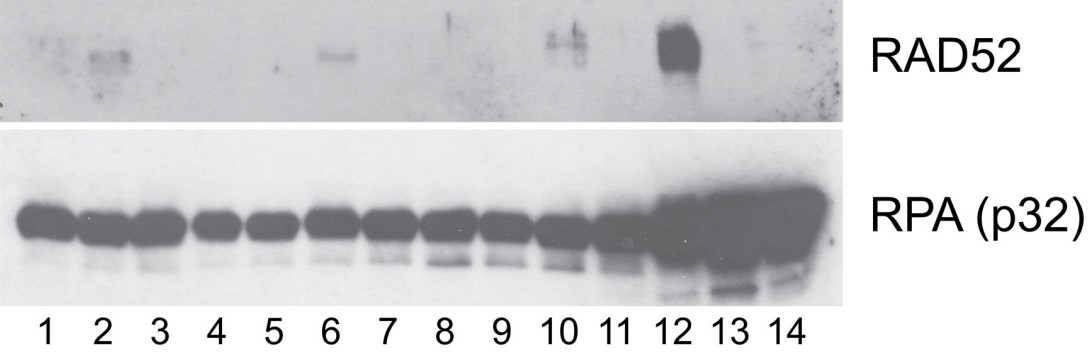

**Fig 9. Mitoxantrone disrupts the RPA:RAD52 PPI in cells. (A)** RAD52 was co-immunoprecipitated with RPA in the fractions 5, 8, and 10 in the control experiment. The elution positions of the molecular weight standards, thyroglobulin (669 kDa), apoferritin (443 kDa), beta-amylase (200 kDa), and bovine serum albumin (66 kDa), are indicated below the western blot. The data indicate that RAD52 and RPA were in complexes with sizes of ~400 kDa and ~200 kDa in the control. **(B)** The treatment of the cells with 3 nM mitoxantrone resulted in the loss of the co-immunoprecipitation of RAD52 with RPA in the large complexex. These data suggest that mitoxantrone disrupted the interaction of RAD52 and RPA in cells.

used extensively as an anti-malarial drug and has emerged recently in "drug repurposing" study as an active agent in subtypes of leukemia, colon and breast cancers [59–61]. It has also been found to inhibit topoisomerase [62]. A surprising case of a lasting complete pathological response to neoadjuvant chemotherapy by a woman presenting with an aggressive stage IV high-grade serous ovarian cancer (HGSC) with metastasis to colon and liver was reported [63]. This unusual response was attributed to receiving quinacrine within a therapeutic regime to treat a separate co-incidental condition of dermatomyositis. Initial assessments to understand the anti-cancer activity of quinacrine was attributed to quinacrine-induced autophagy [64]. It is noteworthy that HGSC cells were used in this study, a subtype of ovarian cancer that has HR-deficiency 50% of the time [65], although the HR status of this particular patient is unclear. Our results revealed RAD52-mediated DNA repair as a potential new target of quinacrine in addition to other known targets.

Topoisomerase II inhibitors exert their action by trapping topoisomerase at DSB ends and inducing DSBs. The accumulation of DSBs by topoisomerase inhibitors is toxic to BRCA-deficient cancer cells. In fact, it has been reported that BRCA-deficient cancer cell lines are sensitive to various topoisomerase inhibitors, including etoposide [66]. In addition to the formation of DSBs, trapped topoisomerases at DSB ends form a highly toxic protein-DNA complex that could prevent DNA repair proteins from accessing the DSB ends to initiate DSB repair. Interestingly, BRCA1, along with MRE11 and CtIP, promotes the removal of the trapped topoisomerases to initiate a resection process in HR repair [67–69]. Therefore, BRCA1-deficient cancer cells show high sensitivity to topoisomerase inhibitors. Our results indicate that the simultaneous inhibition of RAD52-mediated DNA repair with the induction of DSBs appears to exaggerate the cytotoxicity of mitoxantrone and doxorubicin in BRCA1-deficient cancer cell lines.

Resistance to PARPi in BRCA-deficient cancer cells, before or after the treatment with PARPi, is a major issue. Recent studies revealed some mechanistic aspects of the PARPi resistance. Regaining HR and fork protection by inactivation of 53BP1, RIF1, REV7, or a Shieldin, and PTIP, respectively, results in resistance to PARPi in BRCA1-deficient cancer cells [70–79]. Our results demonstrate that our RAD52 inhibitors selectively kill PARPi-resistant BRCA1-deficient cell lines (Figs 2 and 3). Two of the cell lines we used in this study, UWB1-SYr12 and -UWB1-SYr13, obtained resistance to PARPi by acquiring ATR-dependent, but BRCA1-independent HR and fork protection [43]. Thus, RAD52 might play a pivotal role in these ATR-dependent DNA repair pathways. The results also suggest a potential of RAD52 inhibitors to treat a wide range of HR-deficient cancers with or without PARPi-resistance.

## Materials and methods

### Screening of chemical libraries with fluorescence-based protein-protein interaction assay (FluorIA)

A newly developed fluorescence-based assay for detecting protein-protein interactions, FluorIA was used to identify SMIs of RPA:RAD52 PPI. The detailed FluorIA protocol is published elsewhere [39]. Briefly, 384-well plates (Thermo Fisher Scientific, #460518) were coated with purified RPA. Then compounds or controls were added and incubated with freshly prepared GFP-RAD52. RAD52(1–303) without GFP-tag and buffer (including DMSO) only were used as controls. Then the relative fluorescence (RFU) was measured using a POLARstar OPTIMA plate reader (BMG LABTECH) on emptied plates at an excitation/emission (for eGFP) of 485/520 nm at 2500 gain setting.

Three chemical libraries the 355-member SelleckChem Kinase Inhibitor Library, the 1200-member Prestwick Chemical Library, and the 100,000-member ChemBridge library were used for the screening. The screening of the SelleckChem Kinase Inhibitor and the Prestwick libraries were performed at 10 and 100 μM each in duplicate. The screening of the ChemBridge library was conducted at 100 μM. Compounds that inhibit 50% or more of the binding of GFP-RAD52 to RPA compared to controls were considered as hits. The statistical criteria for a hit were fully described in Al-Mugotir et al. [39]. Each hit was verified in triplicate at 10 and 100 μM.

The *in vitro* $EC_{50}$ value of each hit was determined with FluorIA. The chemicals were serially diluted by two-fold increments from 0.030 to 500 μM and analyzed in triplicate with FluorIA. RAD52(1–303) was used as a positive control and buffer with DMSO was used as a negative control. Candidate SMIs with strong inhibition were purchased, and a 1D NMR analysis was carried out to verify their identities. The eleven candidates SMIs were purchased as follows: 5137134, 5246481, 52614577,5316833, 5373030, and 7623574 from Chembridge

HIT2LEAD, 5989499 from Wuxi lab network, 5375405 from Aronis, quinacrine and mitoxantrone from Sigma Aldrich, and doxorubicin from LC labs. D-I03 was purchased from Molport. Their effectiveness in inhibiting the RPA:RAD52 PPI was confirmed after purchase with the FluorIA as described above (S1 Fig).

## Cancer cell lines

The HCC1937 cell line (a gift from Dr. Simon Powell Memorial Sloan Kettering (MSK), New York) was derived from a mammary gland primary ductal carcinoma at the early onset of a tumor. This TNBC patient carried a germline mutation in *BRCA1*, resulting in carboxy-terminal truncated protein and loss of the second *BRCA1* allele. *BRCA1*-restored and empty vector cell lines were also obtained by transfection of a vector containing a full-length human *BRCA1* cDNA or an empty vector, respectively. The HCC1937 cell lines were cultured as described [66].

BRCA2-deficient PE01 cell line (a gift from Dr. Toshiyasu Taniguchi, Tokai University, Japan) were derived from an ovarian cancer patient. BRCA2-proficient "revertant" cell line, C4-2 is one of eight clones derived from PE01 that has acquired resistance to both cisplatin and PARPi due to a secondary mutation that restored the BRCA2 function [62]. PE01 and PE01 C4-2 were grown and maintained in DMEM high glucose (Hyclone) supplemented with 10% Fetal Bovine Serum (FBS, Invitrogen).

UWB1.289 (a gift from Dr. Lee Zou, Harvard Medical School, Boston, MA) were derived from a serous ovarian carcinoma with a germline *BRCA1* mutation and a deletion of the wild-type allele. UWB1.289+BRCA1 is a cell line with stable expression of the wild-type *BRCA1* yielding partial correction of DNA damage response [80]. Two PARP-resistant cell lines, SYr12 and SYr13, were derived from the UWB1.289. The Mammary Epithelial growth media, along with SupplementMix solution for UWB1 cell lines, were obtained from PromoCell GmbH and mixed as described [43, 80].

## Cell viability assay

Cells were seeded $5 \times 10^3$ cells per well in 96-well plates in a 90 μL volume per well one day before treatment. Cells in 96-well plates were incubated with increasing concentrations of the compound or vehicle control (1% DMSO) for 72 hours. Each treatment plate contained nine concentrations of a compound in sextuplicate. Growth inhibition was determined using PrestoBlue reagent (Life Technologies, #A13261). Fluorescent measurements were taken using a Spectramax M5 plate reader (MDS). To assess cells' replication in the absence of compounds as a control, six wells on each plate were treated with vehicle to be read in triplicate at the initial treatment day ($T_0$) and at 72-hour read ($T_{100}$).

## Simple western analysis

For western analysis, each cell line was grown and treated as described above. Protein levels were measured in lysates by size separation with a ProteinSimple PeggySue instrument following the manufacturer's recommendations. Lysates were prepared using the NanoPro Cell Lysis Kit with Bicine/CHAPS (ProteinSimple CBS403) at 0.75 mg/mL concentration when mixed with master mix (containing SDS, DTT and fluorescent molecular weight markers; ProteinSimple PS-FL01-8) and heated for 5 min at 95˚C. Electrophoretic nanocapillary size separation of proteins was performed using default settings. After separation, UV light cross-linked proteins to the capillary wall. Capillaries were then washed and incubated with primary antibodies, washed, and incubated with anti-rabbit secondary antibody conjugated with HRP (ProteinSimple DM-001). Finally, luminol and peroxide were applied to generate

chemoluminescence which was imaged by a CCD camera. Primary antibody concentrations were optimized and were diluted as follows: Abcam anti-RAD52 antibody (ab18263, diluted 1:25), R&D Systems Human Phospho-histone H2AX (S139) antibody (AF2288, diluted 1:100), and Cell Signalling Technology Pan-Actin antibody (#4968, diluted 1:2500).

The Simple western data were processed and analyzed with ProteinSimple Compass software. Peak areas were calculated using dropped lines. Each lysate was analyzed for RAD52 and γH2AX levels three times and for Actin two times (see S2 Fig). Replicate peak areas for RAD52 and γH2AX were averaged, divided by the average Actin peak area of that sample, multiplied by 100 and graphed using SigmaPlot 7.0.

## Statistical analysis

Statistically significant differences in the data were determined using the Analysis of Variance (ANOVA) Single Factor from the Data Analysis package in Microsoft Excel 2016. In Excel, the ANOVA Single Factor is implemented as an F-Test which allows for the simultaneous analysis of multiple groups of data. In our case, we are only comparing two groups for each dosage experiment, so the F-Test is effectively a T-Test. For differences to be significant we required a P Value of less than 0.05 and F must be higher than F critical

## GFP-RAD52 and GFP-RAD51 foci formation

GFP-RAD52 and GFP-RAD51 were stably expressed individually in the HR-restored PE01 C4-2 (S3 Fig). Each cell line was seeded on a coverslip at $10^5$ cells per well in a 24-well plate. After the 24 hour incubation, the cells were pre-incubated with mitoxantrone for 2 hours, then irradiated with X-ray at 15 Gy with the RS-2000 Irradiator. After 2~ 8 hours of the irradiation, the cells were fixed in 10% formaldehyde solution at room temperature for 30 min, followed by 10 min incubation in PBS containing 0.5% Triton X-100. After staining with 1 μg/ml H1399 Hoechst dye solution, each coverslip was mounted on a slide. GFP proteins were detected with a Zeiss 710 Confocal Laser Scanning Microscope, and images were processed using Zeiss ZEN software. A cell contained more than 5 foci was considered as positive.

## Cell-based DSB repair assays

Single strand annealing (SSA) activity and homologous recombination (HR) activity were measured with cell-based DSB repair assays [17]. The U2OS-SA cell line carries the chromosomally integrated SA-GFP reporter and U2OS-DR with chromosomally integrated DR-GFP reporter were used to measure SSA and HR activity, respectively (generous gifts from Dr. Jeremy Stark, Beckman Research Institute of the City of Hope,). U2OS-SA was seeded at $10^5$ cells per well, and U2OS-DR was seeded at $5 \times 10^4$ cells per well in a 12-well plate one day before transfection. U2OS-SA and U2OS-DR were grown and maintained in DMEM high glucose (Hyclone) supplemented with 10% Fetal Bovine Serum (FBS, Invitrogen). RAD52 inhibitors were added to the cells 2 hours before the transfection. The expression plasmid of I-SceI endonuclease (a generous gift from Dr. Maria Jasin at Memorial Sloan Kettering Cancer Center) was transfected to introduce DSBs. Each DSB repair activity was determined by FACS three days after transfection. DSB repair activity was also measured by western blots. The cell lysate was prepared from cells from each well. Expressions of GFP and the I-SceI endonuclease (HA-tagged) were detected with anti-GFP antibody (GenScript A01388-40) and anti-HA antibody (COVANCE MMS-101R) after 12% SDS-PAGE. GAPDH detected by anti-GAPDH antibody (Bethyl Laboratories, A300-639A) was used as a loading control. After normalizing the signals by GAPDH using ImageJ, a ratio of GFP/HA was calculated and used as the DSB repair activity.

## Mitoxantrone effect on RPA:RAD52 complex

U2OS cells ($10^6$ cells) were seeded in a 100 mm dish and grown overnight. The cells were treated with the indicated concentrations of mitoxantrone (MTX) for 24 hours. The cell lysate was prepared in 100 μl of lysis buffer (PBS with 0.5% NP40, 0.5% Triton X100, 1 mM DTT, protease and phosphatase inhibitors, and 10% glycerol). After mixing with 100 μl cold PBS, the lysate was incubated with 100 μl of DEAE sepharose (GE Life Sciences) for 2 hours at 4˚C with rotation. An unbound fraction from DEAE sepharose (~200 μl) was diluted with 200 μl of PBS and half of the mixture was fractionated with Superdex200 (GE Life Sciences) using PBS as elution buffer. Each fraction was incubated with anti-RPA (p32) antibody conjugated with biotin (NeoMarkers, MS-691-B0) at 4˚C overnight, then incubated with Streptavidin-magnetic beads (Promega #Z5481) for one hour at 4˚C with rotation. After washing with PBS, the immuno-precipitated proteins were eluted with 1xSDS loading dye and separated by 10% SDS-PAGE. RAD52 and RPA (p32) were detected by western blotting with anti-RAD52 antibody (LS-C42570, LSBio) and anti-RPA2 antibody (A300-244A, Bethyl Laboratories), respectively.

## Supporting information

**S1 Fig. Validation experiments of eleven hits with the FlourIA.** Sixteen treatment concentrations were used (0.030–500 μM) increasing by two-fold increments of each molecule/drug obtained as a hit in HTS FluorIA. RAD52(1–303) without GFP-tag and buffer (including DMSO) only were used as controls. Percent inhibition was calculated as follows: [(RFU$_{SMI}$—RFU$_{RAD52(1–303)}$) / (RFU$_{DMSO}$—RFU$_{RAD52(1–303)}$)]$^*$100, where RFU is relative fluorescence unit. An average of triplicate measurements was used. The treatment points for quinacrine and doxorubicin beyond 125 μM were showing characteristics of aggregation and were not used.
(TIF)

**S2 Fig. Simple Western analysis of cell lines with and without mitoxantrone treatment.** Data were collected on a nanocapillary PeggySue instrument and displayed with Compass software (Protein Simple) in "lane" mode. **(A)** Results from Abcam anti-RAD52 antibody (ab18263, diluted 1:25). Each sample was run three times. The RAD52 band at 56–58 kDa was confirmed using Phoenix cells (ATCC) overexpressing human RAD52 (not shown). **(B)** Results from R&D Systems Human Phospho-histone H2AX (S139) antibody (AF2288, diluted 1:100). Each sample was run three times and the band at 25 kDa is shown. **(C)** Results from Cell Signaling Technology Pan-Actin antibody (4968, diluted 1:2500). Each sample was run two times and the band at 48 kDa is shown.
(TIF)

**S3 Fig. FACS analysis of cell-based double strand breaks (DSB) repair assay.** DSB repair activity by single strand annealing (SSA) or homologous recombination (HR) in MTX-treated GFP-reporter U2OS-SA and U2OS-DR respectively are measured by FACS analysis. Treated and untreated cells are sorted and repair activity by either pathway is measured by increased green fluorescence (y-axis). Green cells are calculated as a percent from total cells within each contour plot.
(TIF)

**S4 Fig. Representative western blots for detecting DSB repair activities.** SSA (panel A) and HR (panel B) activity were determined by western blotting. The expression levels of I-SceI and GFP were determined by western blots. GADPH was used as a loading control. C: control experiments without I-SceI expression. Lanes 1–3; three independent experiments without

mitoxantrone-treatment, lanes 4–6; three independent experiments with 3 nM mitoxantrone treatment. Cell lysate from each treatment was separated by 12% SDS-PAGE. Two identical samples were analyzed for one set of experiments, and one gel was used for I-SceI expression and the other was used for GFP expression. I-SceI and GFP signals were normalized by the signals of GAPDH in each lane. The repair activity in each lane was expressed as a ratio of normalized GFP/normalized I-SceI.
(TIF)

**S5 Fig. Expression of GFP-RAD52 and GFP-RAD51 in PE01 C4-2 cells.** (A) GFP-RAD52 or GPF-RAD51 were immuno-precipitated by anti-GFP antibody (SCBT B-2), and the immuno-complexes were analyzed on 8% SDS-PAGE followed by the western blots with anti-RAD51 antibody (SCBT H92) and anti-RAD52 antibody (LSBio aa360-375). (B) Expression levels of GFP-RAD52 and GFP-RAD51. Cell lysates from control cells (lane 1), GFP-RAD51 expressing cells (lane 2), and GFP-RAD52 expressing cells (lane 3) were analyzed on 8% SDS-PAGE followed by western blots with anti-GFP antibody (GenScript pAb Rabbit). The arrows indicate GFP-RAD51 (lane 2) and GFP-RAD52 (lane 3). GAPDH was used as a loading control.
(TIF)

**S1 Raw images.**
(PDF)

## Acknowledgments

We thank William Lutz for his technical assistance. We also thank Dr. David Kelly and Amy Wells at the UNMC Fred & Pamela Buffet Cancer Center Molecular Biology/HTS Core Facility for access to the chemical libraries and instrumentation. We also thank the University of Nebraska Medical Center Advanced Microscopy Core Facility, Flow Cytometry Research Facility, and Biological Irradiator Core Facility.

## Author Contributions

**Conceptualization:** Amarnath Natarajan, Tadayoshi Bessho, Gloria E. O. Borgstahl.

**Funding acquisition:** Tadayoshi Bessho, Gloria E. O. Borgstahl.

**Investigation:** Mona Al-Mugotir, Joseph George, Mika Bessho, Dhananjaya Pal, Lucas Struble, Carol Kolar, Sandeep Rana, Tadayoshi Bessho.

**Methodology:** Tadayoshi Bessho.

**Supervision:** Gloria E. O. Borgstahl.

**Validation:** Sandeep Rana.

**Visualization:** Tadayoshi Bessho.

**Writing – original draft:** Mona Al-Mugotir, Gloria E. O. Borgstahl.

**Writing – review & editing:** Jeffrey J. Lovelace, Tadayoshi Bessho, Gloria E. O. Borgstahl.

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
