## [Decision Letter · Decision Letter 0]

8 Jul 2020

PONE-D-20-16153

Selective killing of homologous recombination-deficient cancer cell lines by inhibitors of the RPA:RAD52 protein-protein interaction

PLOS ONE

Dear Dr. Borgstahl,

Thank you for submitting your manuscript to PLOS ONE. After careful consideration, we feel that it has merit but does not fully meet PLOS ONE’s publication criteria as it currently stands. Therefore, we invite you to submit a revised version of the manuscript that addresses the points raised during the review process.

We look forward to receiving your revised manuscript.

Kind regards,

Alvaro Galli

Academic Editor

PLOS ONE

Journal Requirements:

2. Please ensure you have thoroughly discussed any potential limitations of this study within the Discussion section.

Reviewers' comments:

Reviewer's Responses to Questions

**Comments to the Author**

1. Is the manuscript technically sound, and do the data support the conclusions?

Reviewer #1: Yes

Reviewer #2: Partly

2. Has the statistical analysis been performed appropriately and rigorously? 

Reviewer #1: No

Reviewer #2: I Don't Know

3. Have the authors made all data underlying the findings in their manuscript fully available?

Reviewer #1: Yes

Reviewer #2: Yes

4. Is the manuscript presented in an intelligible fashion and written in standard English?

Reviewer #1: Yes

Reviewer #2: Yes

5. Review Comments to the Author

Reviewer #1: This is a very well written manuscript with novel and exciting data on using FDA approved drugs for targeted therapy of BRCA-1/2 cancers by specifically target the RPA:RAD52 protein interaction.

Comments:

1. Figures need to be of higher quality/resolution/contrast

2. Identification of the axes in some figure need to be more clear for example figure 1 y axes "% inhibition of " what?) while it is in the text and the legend it will be more clear if it was on the axes it self

3. Most of the supplemental data should be actually in the main manuscript that S2-6 figures as the some claims and conclusions in the manuscript can not stand without the data in these figures. The data in the manuscript should stand by it self even without the supporting material.

4. Indicate significance in each figure where applied ( use * or similar marking)

5. Unpaired student t test is not an adequate statistical tool for some the data that have more than three variables authors need to do ANOVA with honesty testing.

6 Minor spelling and grammar issues that can be easily fixed

Overall all the manuscript have excellent experimental design, adequate controls and pass the scientific rigor and it would be outstanding with these minor suggestion an comments.

Reviewer #2: In current manuscript, authors identified three FDA-approved drugs Mitoxantrone, doxorubicin and quinacrine as the inhibitors of the RPA:RAD PPI by high throughput screening from chemical libraries. They demonstrated that these drugs could induce synthetic lethality in BRCA deficient cancer cell lines in vitro. Overall this is a preliminary study. Some points need to be addressed.

1. To address that RAD52 PPI inhibitors induce synthetic lethality in

HR-deficient cancer cell lines, in Fig2-4, it should show the RAD52 expression in the tested cell lines before and after treatment. RAD52 is DNA Repair Protein. It also would be better to have the evidence that these drugs induced DNA damage in the tested cell lines, such as markers γH2AX.

2. Fig3A, X axis missed concentration label “uM”.

3. Fig 5C, is it possible to show western blot original images and FACS image?

4. FigS3A, B, it was not clear which cell line was treated. Can authors label them?

5. Fig S4, Western blot didn’t show loading control although authors claimed there were GAPDH controls in the experiments. (Not sure it was the image loading problem, it seemed Fig S4 missed left panel?)

6. Labels in figure make people confuse. Fig S5, what does mean “1,2,3,4,5,6”? Was drug dose? FigS4, S5, was “c” control? Scel expressed in Fig S4”c” but not in Fig S5 “c”. What was the difference of the “C”?

6. PLOS authors have the option to publish the peer review history of their article (what does this mean?). If published, this will include your full peer review and any attached files.

Reviewer #1: No

Reviewer #2: No

---

## [Author Response · Author response to Decision Letter 0]

29 Jan 2021

The reviewers’ comments are summarized below and followed by our response in purple italics. Places in the manuscript that were modified were highlighted in gray. 

Reviewer #1: This is a very well written manuscript with novel and exciting data on using FDA approved drugs for targeted therapy of BRCA-1/2 cancers by specifically target the RPA:RAD52 protein interaction.

Comments:

1. Figures need to be of higher quality/resolution/contrast

The resolution of all figures has been improved.

2. Identification of the axes in some figure need to be more clear for example figure 1 y axes "% inhibition of " what?) While it is in the text and the legend it will be more clear if it was on the axes itself

The y-axis for Figure 1B was revised to indicate what is being inhibited (RPA:RAD52 complex). 

3. Most of the supplemental data should be actually in the main manuscript that S2-6 figures as the some claims and conclusions in the manuscript cannot stand without the data in these figures. The data in the manuscript should stand by itself even without the supporting material.

Although the data in the supplement support the manuscript conclusions, we consider these data as preliminary and this is an active area of research in the lab. Therefore, we believe the supplementary section is the appropriate place so they do not detract from the focus of the manuscript. 

4. Indicate significance in each figure where applied (use * or similar marking)

The data in Figures 2, 3, and 4 were re-evaluated using ANOVA honesty testing as described in the edited method section of the manuscript and an asterisk sign was indicated where significance was found. 

5. Unpaired student t test is not an adequate statistical tool for some the data that have more than three variables authors need to do ANOVA with honesty testing.

ANOVA with honesty testing was used to re-evaluate the statistical significance of the data as described in the revised method section.

6. Minor spelling and grammar issues that can be easily fixed

The manuscript was edited for any spelling or grammar issues

Overall all the manuscript have excellent experimental design, adequate controls and pass the scientific rigor and it would be outstanding with these minor suggestion an comments.

Reviewer #2: In current manuscript, authors identified three FDA-approved drugs mitoxantrone, doxorubicin and quinacrine as the inhibitors of the RPA:RAD PPI by high throughput screening from chemical libraries. They demonstrated that these drugs could induce synthetic lethality in BRCA deficient cancer cell lines in vitro. Overall this is a preliminary study. Some points need to be addressed.

1. To address that RAD52 PPI inhibitors induce synthetic lethality in HR-deficient cancer cell lines, in Fig2-4, it should show the RAD52 expression in the tested cell lines before and after treatment. RAD52 is DNA Repair Protein. It also would be better to have the evidence that these drugs induced DNA damage in the tested cell lines, such as markers γH2AX.

Thank you, this was an excellent idea and the resulting data proved to be very informative. All cell lines were grown and cells were harvested without treatment and treated with doses of mitoxantrone near their EC50 values. The level of DNA damage induced (�-H2AX) and RAD52 expression was measured by western analysis as detailed in the method section. The western data were measured in triplicate, normalized with actin and graphed (average and standard deviation) with significance calculated using ANOVA: Single factor in Excel. These graphs were included as a new figure (Fig. 5) and so the figures after that were renumbered. Individual data measurements were included in the supplementary section using the lane representation (S2 Fig.). Western data were measured using a Protein Simple Peggy Sue instrument and analyzed using the associated Compass for Simple Western software.

2. Fig3A, X axis missed concentration label “uM.

Corrected.

3. Fig 5 (now Fig 6), is it possible to show western blot original images (parts and FACS image?

Western images are in Fig S6. The detailed FACS analysis from our core facility was added to Fig S6.

4. FigS3A, B, it was not clear which cell line was treated. Can authors label them?

Corrected.

5. Fig S4, Western blot didn’t show loading control although authors claimed there were GAPDH controls in the experiments. (Not sure it was the image loading problem, it seemed Fig S4 missed left panel?)

We did not describe that we used GAPDH as a loading control. We used p70 subunit of RPA to normalize the expression of SceI. All information is given in the legend.

6. Labels in figure make people confuse. Fig S5, what does mean “1,2,3,4,5,6”? Was drug dose? FigS4, S5, was “c” control? Scel expressed in Fig S4”c” but not in Fig S5 “c”. What was the difference of the “C”?

We have modified the labeling of Fig. S4 so the meaning is clearer. Now, in both figures, C stands for no SceI control. The 1,2,3,4,5,6 in Fig. S5 are simply lane numbers as described in the figure legend. We have revised the figure legends for both Figs. S4 and S5 for clarity.

---

## [Decision Letter · Decision Letter 1]

11 Feb 2021

PONE-D-20-16153R1

Selective killing of homologous recombination-deficient cancer cell lines by inhibitors of the RPA:RAD52 protein-protein interaction

PLOS ONE

Dear Dr. Borgstahl,

Thank you for submitting your manuscript to PLOS ONE. After careful consideration, we feel that it has merit but does not fully meet PLOS ONE’s publication criteria as it currently stands. Therefore, we invite you to submit a revised version of the manuscript that addresses the points raised during the review process.

The reviewer is asking why some results are included in the supplementary materials and not in the manuscript figure; these results are discussed in the text and should be shown in the main text. I believe that that fugure should be shown in the manuscript properly.

We look forward to receiving your revised manuscript.

Kind regards,

Alvaro Galli

Academic Editor

PLOS ONE

Reviewers' comments:

Reviewer's Responses to Questions

**Comments to the Author**

1. If the authors have adequately addressed your comments raised in a previous round of review and you feel that this manuscript is now acceptable for publication, you may indicate that here to bypass the “Comments to the Author” section, enter your conflict of interest statement in the “Confidential to Editor” section, and submit your "Accept" recommendation.

Reviewer #1: All comments have been addressed

2. Is the manuscript technically sound, and do the data support the conclusions?

Reviewer #1: Partly

3. Has the statistical analysis been performed appropriately and rigorously? 

Reviewer #1: Yes

4. Have the authors made all data underlying the findings in their manuscript fully available?

Reviewer #1: Yes

5. Is the manuscript presented in an intelligible fashion and written in standard English?

Reviewer #1: Yes

6. Review Comments to the Author

Reviewer #1: The authors have addressed all my comment. I still stand by my comment that supplementary data presented was used in the claims and conclusions of the study and the manuscript cannot stand without the data in these figures. I am surprised that the authors characterized these data as preliminary and not complete and still under investigation. The authors have the option to remove the claims and the conclusions related to these supplementary (preliminary) data from the main manuscript. The figure are still have poor resolution, but I think that not the fault of the authors just technology not working appropriately!

7. PLOS authors have the option to publish the peer review history of their article (what does this mean?). If published, this will include your full peer review and any attached files.

Reviewer #1: No

---

## [Author Response · Author response to Decision Letter 1]

5 Mar 2021

We have implemented the suggestions from the reviewer by moving three figures from the supplement into the manuscript for the data that was directly cited in the text. There is a total of 9 figures in the main manuscript. The figures that remain in the supplement are only cited in the figure legends as they contain the original data that is summarized in graphs in the main figure. Some methods were moved from figure legends (previously in the supplement) to the methods section. These changes are highlighted in grey in the “Revised manuscript with Track Changes” document.

Concerning the resolution of the figures, we have uploaded high resolution figures. We are not sure why the figures the reviewer sees appear to be low resolution.

---

## [Decision Letter · Decision Letter 2]

9 Mar 2021

Selective killing of homologous recombination-deficient cancer cell lines by inhibitors of the RPA:RAD52 protein-protein interaction

PONE-D-20-16153R2

Dear Dr. Borgstahl,

We’re pleased to inform you that your manuscript has been judged scientifically suitable for publication and will be formally accepted for publication once it meets all outstanding technical requirements.

Kind regards,

Alvaro Galli

Academic Editor

PLOS ONE

Additional Editor Comments (optional):

Reviewers' comments:

Reviewer's Responses to Questions

**Comments to the Author**

1. If the authors have adequately addressed your comments raised in a previous round of review and you feel that this manuscript is now acceptable for publication, you may indicate that here to bypass the “Comments to the Author” section, enter your conflict of interest statement in the “Confidential to Editor” section, and submit your "Accept" recommendation.

Reviewer #1: All comments have been addressed

2. Is the manuscript technically sound, and do the data support the conclusions?

Reviewer #1: (No Response)

3. Has the statistical analysis been performed appropriately and rigorously? 

Reviewer #1: Yes

4. Have the authors made all data underlying the findings in their manuscript fully available?

Reviewer #1: Yes

5. Is the manuscript presented in an intelligible fashion and written in standard English?

Reviewer #1: Yes

6. Review Comments to the Author

Reviewer #1: The authors have addressed all of my comments. I think the manuscript in this revised version is highly enhanced and will attract a lot of attention and citations.

7. PLOS authors have the option to publish the peer review history of their article (what does this mean?). If published, this will include your full peer review and any attached files.

Reviewer #1: No

---

## [Editor Report · Acceptance letter]

18 Mar 2021

PONE-D-20-16153R2 

Selective killing of homologous recombination-deficient cancer cell lines by inhibitors of the RPA:RAD52 protein-protein interaction 

Dear Dr. Borgstahl:

I'm pleased to inform you that your manuscript has been deemed suitable for publication in PLOS ONE. Congratulations! Your manuscript is now with our production department. 

Kind regards, 

on behalf of

Dr. Alvaro Galli 

Academic Editor

PLOS ONE